# Collaborative Symmetricity Exploitation for Offline Learning of Hardware Design Solver

**Haeyeon Kim\*, Minsu Kim\*, Joungho Kim, and Jinkyoo Park**
(\* for equal contribution)
Korea Advanced Institute of Science and Technology (KAIST)
{haeyeonkim, min-su, joungho, jinkyoo.park}@kaist.ac.kr

## Abstract

This paper proposes *collaborative symmetricity exploitation* (CSE) framework to train a solver for the decoupling capacitor placement problem (DPP), one of the significant hardware design problems. Due to the sequentially coupled multi-level property of the hardware design process, the design condition of DPP changes depending on the design of higher-level problems. Also, the online evaluation of real-world electrical performance through simulation is extremely costly. Thus, we propose the CSE framework that allows data-efficient offline learning of a DPP solver (i.e., contextualized policy) with high generalization capability over changing task conditions. Leveraging the symmetricity for offline learning of hardware design solver increases data-efficiency by reducing the solution space and improves generalization capability by capturing the invariant nature present regardless of changing conditions. Extensive experiments verified that CSE with zero-shot inference outperforms the neural baselines and iterative conventional design methods on the DPP benchmark. Furthermore, CSE greatly outperformed the expert method used to generate the offline data for training.

## 1 Introduction

Many studies have shown that deep reinforcement learning (DRL) is promising in various tasks in modern chip design; chip placement [1, 2], routing [3, 4], circuit design [5], logic synthesis [6, 7] and bi-level hardware optimization [8]. However, most of them do not take the following into consideration. (a) Online simulators for hardware are usually time intensive and inaccurate; thus, learning with existing offline data by experts is more reliable. Since there exists a limited number of offline hardware data, a data-efficient learning scheme is necessary. (b) Hardware design is composed of electrically coupled multi-level tasks where task conditions are determined by the design of higher-level tasks; thus, a solver (i.e., contextualized policy conditioned by higher-level tasks) with high generalization capability to adapt to varying task conditions is necessary.

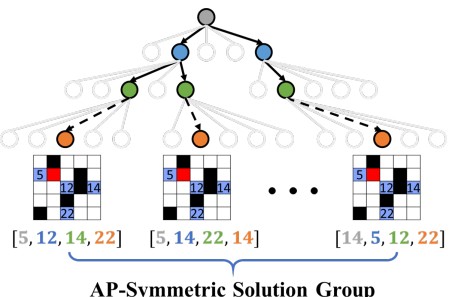

Figure 1: Conventional sequential decision-making method's heterogeneous trajectories from AP-Equvariant solution group.

We observed that leveraging the symmetricity in placement problems can effectively improve both data-efficiency of training and generalization capability of trained solver over task condition variation. As shown in Fig. 1, the conventional sequential decision-making schemes for placement problems [9, 1, 8] search over a heterogeneous trajectory, referring to a group of $K!$ action-permutation

Offline Reinforcement Learning Workshop at Neural Information Processing Systems, 2022.

(AP)-symmetric solution trajectories, where $K$ is the number of actions. If each group of AP-symmetric solution trajectories can be identified as a single merged trajectory, the search space can be dramatically reduced, giving two major advantages. First, inducing symmetricity allows data-efficient training of solver with a limited number of offline expert data because the amount of training data is proportional to the policy's search space. Second, inducing symmetricity improves generalization capability of the solver to varying task conditions because symmetricity is an invariant nature shared among tasks regardless of conditions.

To this end, we devised *collaborative symmetricity exploitation* (CSE) framework, a simple but effective method to induce AP-symmetricity with two collaborative learning schemes: expert exploitation and self-exploitation. To further improve the generalization capability of the trained solver, we also devised a target problem-specific neural architecture by modifying the attention model (AM) [10] with two problem-specific context neural networks.

**Related Works.** Several studies to leveraged the symmetricity in solution space. [11] suggested the policy optimization for multiple optima (POMO) scheme to leverage the traveling salesman problem (TSP)'s solution symmetricity, the cyclic property that identical solution can be expressed as $N$ heterogeneous trajectories by permuting initially visited node. [12] proposed the symmetric neural combinatorial optimization (Sym-NCO) method, a general-purpose symmetric learning method. [13] proposed a generative flow net (GFlowNet) to train policy distribution proportional to reward distribution $\pi \propto R$ considering solution symmetricity. While POMO [11] and Sym-NCO [12] leverage DRL, CSE focuses on offline imitation learning. Though GFlowNet [13] can be trained in a fully offline manner, it is not yet designed for training a contextualized policy. CSE is an offline symmetricity learning method to train contextualized policy.

## 2 Decap Placement Problem (DPP) Formulation

This paper seeks to solve the decap placement problem (DPP), one of the essential hardware design problems. Decoupling capacitor (decap) is a hardware component that reduces power noise along with the power distribution network (PDN) of hardware devices and improves the power integrity (PI). The goal of DPP is to optimally place a pre-defined number ($K$) of decaps on a ($N_{row} \times N_{col}$)-sized target PDN, given two conditions determined by higher-level tasks; keep-out regions and a probing port location [14]. Keep-out regions are action-restricted areas where decaps cannot be placed as a design constraint. Probing port is the target chip/logic block location where the objective, power integrity (PI), is evaluated. Generally, the more decaps are placed, the more reliable the power supply is. However, adding more decaps requires more space and is costly. Thus, finding an optimal placement of decaps is essential in terms of hardware performance and cost/space-saving.

**DPP Benchmark PDN and Decap Specifications.** The PDN model for verification has ($N_{row} \times N_{col}$) = ($10 \times 10$) grids over 201 frequency points linearly distributed between 200MHz and 20GHz, which gives $100 \times 100 \times 201 \approx 2M$ impedances to be evaluated. Out of the $N_{row} \times N_{col}$ ports, one is assigned as a probing port and 0 to 15 ports are assigned as keep-out ports (see Appendix A.4). The RLGC electrical parameters of PDN and decap in the benchmark are shown in Appendix A.2.

**Objective Function of DPP** The objective of DPP is evaluated by power integrity (PI) simulation that computes the level of impedance suppression over a specified frequency domain:

$$\mathcal{J} := \sum_{f \in F} (Z_{initial}(f) - Z_{final}(f)) \cdot \frac{1\text{GHz}}{f} \tag{1}$$

where $Z_{initial}$ and $Z_{final}$ are the initial and final impedance at the frequency $f$ before and after placing decaps, respectively. $F$ is the set of specified frequency points. The more impedance is suppressed, the better the power integrity and the higher the performance score. Remark that DPP cannot be formulated as a conventional mixed-integer linear programming (MILP)-based combinatorial optimization because PI performance can not be formulated as a closed analytical form but can only be measured or simulated.

### 2.1 Markov Decision Process (MDP)

As shown in Fig. 2, the procedure for solving DPP is modeled as a Markov decision process(MDP). The *task-condtioned* PDN is represented as a set of three-dimensional feature vectors

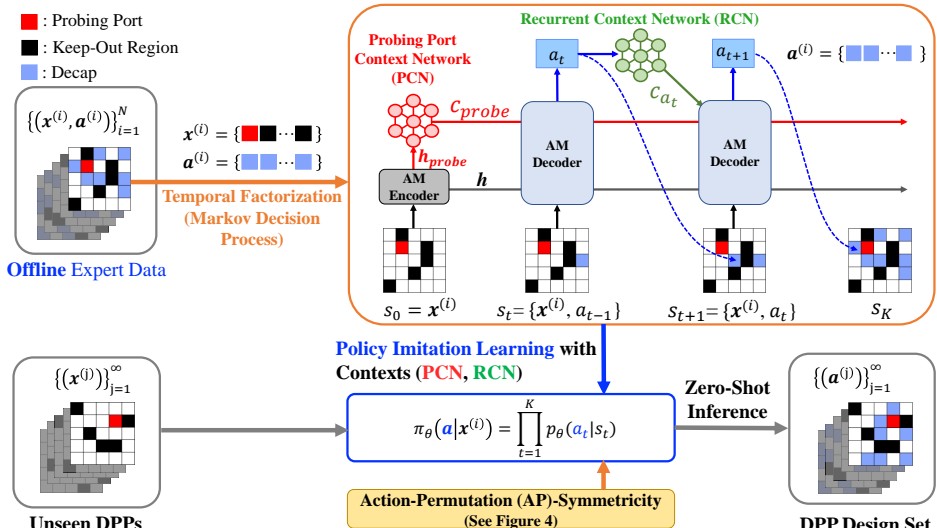

Figure 2: Overall pipeline of DPP contextualized policy parameterization process.

$\boldsymbol{x} = \{\boldsymbol{x}_i\}_{i=1}^{N_{row} \times N_{col}}$, where each grid (i.e., port) on PDN is represented as $\boldsymbol{x}_i = (x_i, y_i, c_i)$, in which $x_i, y_i$ indicate 2D coordinates of location, $c_i$ indicates the condition of port whether it belongs to a probing port $I_{probe}$ ($c_i = 2$), keep-out regions $I_{keepout}$ ($c_i = 1$), or the rest decap allowed ports $I_{allowed}$ ($c_i = 0$). See Appendix A.3.

**State** $\boldsymbol{s}_t$ contains task-condition $\boldsymbol{x}$ and previous selected actions: $\boldsymbol{s}_t = \{\boldsymbol{x}, \boldsymbol{a}_{1:t-1}\}$.

**Action** $a_t \in \{1, ..., N_{row} \times N_{col}\} \setminus \boldsymbol{s}_{t-1}$ is the allocation of a decap to one of the available ports on PDN. The concatenation of sequentially selected actions $\boldsymbol{a} = a_{1:K}$ becomes the final *solution*.

**Policy** $\pi_\theta(\boldsymbol{a}|\boldsymbol{x})$ is the probability of producing a specific solution $\boldsymbol{a} = \boldsymbol{a}_{1:K}$, given task-condition $\boldsymbol{x}$, and is factorized as:

$$\pi_\theta(\boldsymbol{a}|\boldsymbol{x}) = \prod_{t=1}^{K} p_\theta(a_t|\boldsymbol{s_t}), \tag{2}$$

where $p_\theta(a_t|\boldsymbol{s_t})$ is the segmented one-step action policy parameterized by the neural network.

The objective of DPP is to find the optimal parameter $\theta^*$ of the policy $\pi_\theta(\cdot|\boldsymbol{x})$ as:

$$\theta^* = \arg\max_\theta \mathbb{E}_{\boldsymbol{x} \sim \rho} \mathbb{E}_{\boldsymbol{a} \sim \pi_\theta(\cdot|\boldsymbol{x})} \big[ \mathcal{J}(\boldsymbol{a}) \big], \tag{3}$$

where $\rho$ is the probability distribution for varying *task-condition* $\boldsymbol{x}$ and $\mathcal{J}$ is objective function. Once the task $\boldsymbol{x}$ is specified by $\rho$, the state-action space with complexity of $\binom{N_{row} \times N_{col} - 1 - |I_{keepout}|}{K}$ is determined. Thus, an efficient policy $\pi_\theta(\boldsymbol{a}|\boldsymbol{x})$ should capture the contextual features among varying task conditions $\boldsymbol{x}$.

## 3 Methodology

The symmetricity found in placement problems is the action-permutation (AP)-symmetricity, the order of placement does not affect the design performance. Let $t_i$ a permutation of an action sequence $\{1, ..., K\}$, where $K$ is the length of the action sequence. We then define the AP-transformation $T_{AP} = \{t_i\}_{i=1}^{K!}$ as a set of all possible permutations. The AP-symmetricity of DPP is induced to the learned solver through the AP-transformation $T_{AP}$.

### 3.1 Collaborative Symmetricity Exploitation (CSE) Framework

The CSE framework was designed to induce the AP-symmetricity to the trained model to improve the generalization capability and to allow data-efficient training. To train a contextualized policy with a limited number of expert data, we designed the CSE loss term $\mathcal{L}$ consisting of expert exploitation loss $\mathcal{L}_{Expert}$ and self-exploitation loss $\mathcal{L}_{Self}$ as follows:

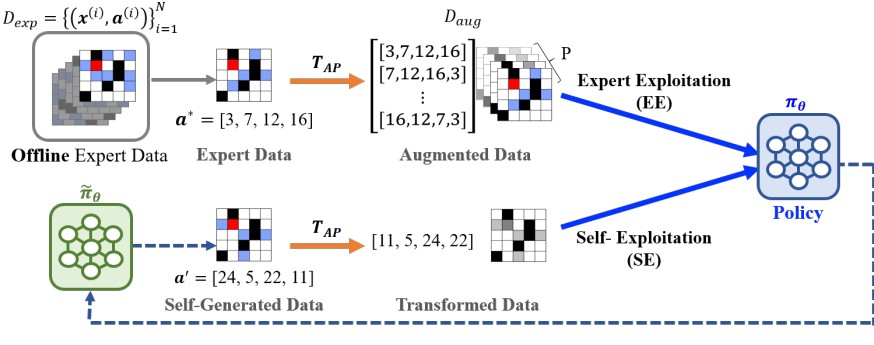

Figure 3: Illustration of collaborative symmetricity exploitation (CSE) process.

$$\mathcal{L} := \mathcal{L}_{Expert} + \lambda\mathcal{L}_{Self} \tag{4}$$

$$\mathcal{L}_{Expert} = -\mathbb{E}_{\boldsymbol{a}^*,\boldsymbol{x}\sim D_{aug}}[log\pi_\theta(\boldsymbol{a}^*|\boldsymbol{x})] \tag{5}$$

$$\mathcal{L}_{Self} = \mathbb{E}_{\boldsymbol{x}\sim\mathcal{U}(\mathcal{X})}\mathbb{E}_{\boldsymbol{a}'\sim\pi_{\tilde{\theta}}(\cdot|\boldsymbol{x})}\mathbb{E}_{t\sim\mathcal{U}(T_{AP})}[||\pi_{\tilde{\theta}}(\boldsymbol{a}'|\boldsymbol{x}) - \pi_\theta(t(\boldsymbol{a}')|\boldsymbol{x})||_1] \tag{6}$$

**Expert Exploitation.** Expert exploitation trains a high-quality symmetric contextualized policy for various task-conditions $x$ by leveraging the offline expert data $a^*$ with $T_{AP}$ that transforms the existing offline expert data $a^*$ for $P$ times to *augment* the offline expert dataset $D_{exp} = \{(x^{(i)}, a^{(i)*})\}_{i=1}^N$. Specifically, we randomly choose $\{t_1, ..., t_P\} \subset T_{AP}$ to generate $D_{aug} = \{(x^{(i)}, a^{(i)*}), (x^{(i)}, t_1(a^{(i)*})), ..., (x^{(i)}, t_P(a^{(i)*}))\}_{i=1}^N$. Then, $\mathcal{L}_{Expert}$ is expressed as a *teacher-forcing* imitation learning scheme with the augmented expert dataset $D_{aug}$.

**Self-Exploitation.** While $D_{aug}$ only contains expert quality data, self-exploitation involves self-generated data, whose quality is poor at the beginning but improves over the phase of training so that induces the AP-symmetricity in a wider action space to achieve greater generalization capability. The proposed $\mathcal{L}_{Self}$ has three probability distributions. First, the $\mathcal{U}(\mathcal{X})$ indicates a uniform distribution of task-condition set $\mathcal{X}$ which helps to learn a contextualized policy capable of adaptation to task variations. Second, the $\pi_{\tilde{\theta}}$ is a fixed copy of the current policy during the training, which samples a pseudo label data $\boldsymbol{a}'$ for task-condition $\boldsymbol{x}$. Lastly, the $\mathcal{U}(T_{AP})$ is a uniform distribution that samples permutation $t$ from $T_{AP}$. Then, $\mathcal{L}_{Self}$ is expressed as an expectation of $L_1$ loss between the probability to generate $\boldsymbol{a}'$ and the probability to generate $t(\boldsymbol{a}')$. By minimizing $L_{self}$, we can enforce the probabilities of generating $a'$ and $t(a')$ to be identical, thus imposing AP-symmetricity directly into the current policy.

### 3.2 Contextual Attention Model

To further improve the generalization capability of the DPP solver, we modified the attention model (AM) [10] and termed *contextual attention model*. As shown in Fig. 2, the decision-making procedure consists of two newly devised context neural networks; (1) encoder capturing initial design conditions while contextualizing the probing port through the probing port context network (PCN) and (2) decoder sequentially allocating decaps on PDN while contextualizing the previous partial solution through the recurrent context network (RCN). See Appendix C.

## 4   Experimental Results

**Offline Expert Data Collection.** We synthetically generated offline expert data using genetic algorithm (GA) for this study. The number of iterations done for collecting a single data is represented as $M$. We used GA$\{M = 100\}$ to collect the offline expert data. In addition, we denote $N$ as the number of offline expert data used for training the CSE.

**Hyperparameters.** For training, we set $P(= 3)$, the number of AP-transformed data per offline data. We set the distribution $\rho$, described in Section 2.1, as a uniform distribution for training. We used $N = 2000$ offline expert data for training CSE and imitation learning-based baselines. We trained our model with batch size 100 for $N < 200$ and batch size $1,000$ for $N = 1,000$ and $2,000$. We trained for a maximum of 200 epochs for each model and used the model with the best validation score to evaluate performance. See Appendix D for details.

Table 1: Performance evaluation with the average score of 100 PDN cases (the higher the better).

| Method | Method Type | PI Simulation ($M$) | Avg. Score |
|---|---|---|---|
| Random Search | Online Search | 10,000 | 12.70 |
| Genetic Algorithm (*expert policy*) | Online Search | 100 | 12.56 |
| Genetic Algorithm | Online Search | 500 | 12.79 |
| AM-RL [15] | Pretrained | 1 | 11.71 |
| Arb-RL [14] | Pretrained | 1 | 9.60 |
| AM [15]-IL | Pretrained | 1 | 12.06 |
| Arb [14]-IL | Pretrained | 1 | 10.80 |
| **CSE** (*ours*) | Pretrained | 1 | **12.88** |

**Baselines for Comparison.** For search heuristic baseline methods, we implemented random search and genetic algorithm that require a large number of simulations (i.e., $M >= 100$) for each problem. Also, we reported two RL baselines, AM-RL [15] and Arb-RL [14] and two IL baselines, AM-IL and Arb-IL, which are modified AM-RL and Arb-RL with imitation learning. Implementation details of the baselines are provided in Appendix D.2 and Appendix D.3.

## 4.1 Generalization Capability Evaluation

To verify the generalization capability of the trained solver, each method is given the same unseen 100 DPPs and the average performance score was measured, after allocating $K = 20$ decaps on each. We made sure test data, validation data and training data did not overlap.

As shown in Table 1, our CSE significantly outperformed all baselines. While online search methods generally achieved high performance as they require a large number of searching iterations $M$ per problem, the learning-based baselines, once trained, only required a single simulation $M = 1$ to measure the performance. For training, when the number of costly simulations was limited, RL-based methods showed poorer generalization capability than their imitation learning versions due to inefficiency in exploring over extremely large combinatorial action space of DPP. We believe that imitation learning approach has greater exploration capability with the help of expert policy thus able to achieve higher performance with a limited simulation budget (see Appendix D.2). Note that if we have an infinite budget for simulation, DRL could achieve greater performance with a sufficient learning loop. Among the imitation learning approaches trained with the same number of offline expert data ($N = 2,000$), CSE showed the highest performance.

The CSE policy trained with offline expert data generated by GA{100} outperformed GA{500} with zero-shot inference. The CSE policy trained with low-quality offline expert data produced higher-quality designs. We believe this was possible as we trained a factorized form of policy that does not predict labels in a single step but produced a solution through a serial iterative roll-out process, during which a good strategy for placing decaps can be identified.

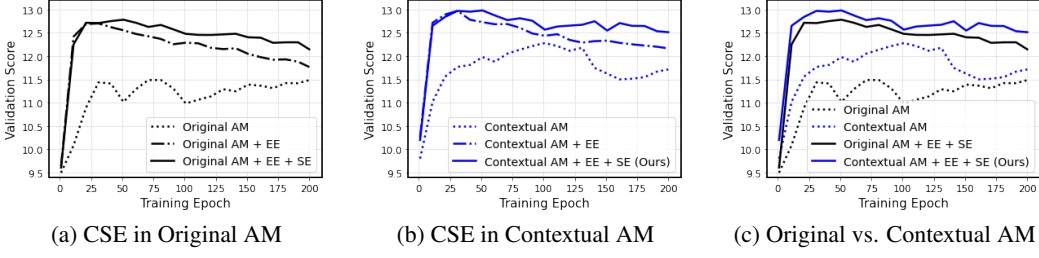

(a) CSE in Original AM     (b) CSE in Contextual AM     (c) Original vs. Contextual AM

Figure 4: Ablation study on CSE components

**Ablation Study.** We conducted ablation studies on CSE components and context neural networks with sparse offline data ($N = 100$). We ablated the effectiveness of expert exploitation (EE) and self-exploitation (SE) in two policy networks: original AM (AM-IL baseline) and contextual AM (ours). Each component of CSE supported increasing generalization capability in both policy networks. The contextual AM with newly devised context neural networks outperformed the original AM.

## 4.2 Offline Data Efficiency Evaluation

$N$ is the number of offline expert data generated by the expert method, GA $\{M = 100\}$. We ablated $N \in \{100, 500, 1000, 2000\}$ with fixed $P = 3$ and compared to the AM-IL baseline. As shown in Fig. 5, CSE outperformed AM-IL baseline in all $N$ variation and CSE with $N = 100$ achieved a greater score than AM-IL with $N = 2000$. In addition, the performance of AM-IL saturates at $N > 500$ while the performance of CSE continuously increases with $N$.

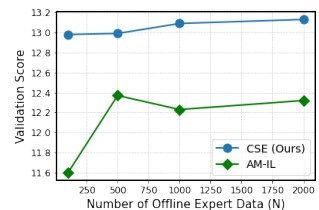

Figure 5: Offline data-efficiency evaluation

## 5 Conclusion

This paper proposed the *collaborative symmetricity exploitation* (CSE) framework for training a contextualized policy (i.e., solver) of placement tasks in an offline manner. The CSE was applied to decap placement problem (DPP) and achieved the most promising performance among all baseline methods. The CSE is a general purpose offline learning scheme for placement tasks that can be further applied to other hardware placement tasks including chip placement, ball grid array (BGA) placement, and via placement.

**Acknowledgement**
This research was supported by the BK21 FOUR project. We would also like to acknowledge the technical support of Ansys Korea.

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

# A  DPP Electrical Modeling and Problem Definition

This section provides electrical modeling details of PDN and decap models used for verification of CDML in DPP. Note that these electrical models can be substituted by those of your interest. There are three methods to extract PDN and decap models that are also used for objective evaluation; 3D EM simulation tool, ADS circuit simulation tool, and unit-cell segmentation method. For each method, there exists a trade-off between time complexity and accuracy. See Table 2. Out of the three methods, we used the unit-cell segmentation method for a benchmark. Simulation time was evaluated using the same PDN model on Intel i7. Note that simulation time depends on the size and complexity of the PDN model.

Table 2: Time Taken for an Objective Evaluation of a PDN model described in Appendix A.2

| Simulation Method | Time Taken |
| --- | --- |
| EM Simulation Tool | $\approx$10 hours |
| ADS Circuit Simulation Tool | 23.58 sec |

## A.1  Domain Perspective Decap Placement Problem

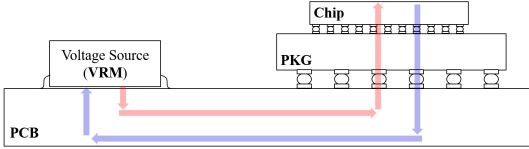

(a) An example of hierarchical power distribution network (PDN).

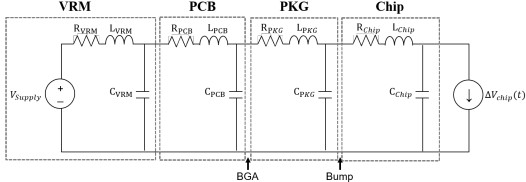

(b) Electrical circuit model of the hierarchical PDN in (a).

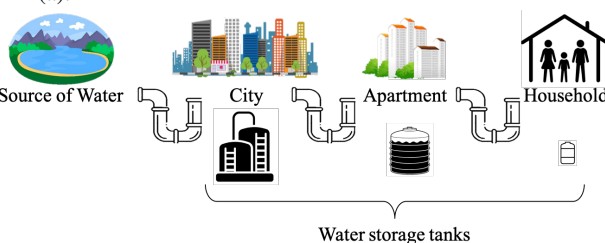

(c) Water supply chain from the source to household.

Figure 6: Illustration of Hierarchical Power Distribution Network (PDN) analogous to Water Supply Chain.

The development of AI has led to an increased demand for high-performance computing systems. High-performance computing systems not only require precise design of hardware chips such as CPU, GPU and DRAM, but also require stable delivery of power to the operating integrated circuits. Power delivery has become a huge technical bottleneck of hardware devices due to the continuously decreasing supply voltage margin along with the technology shrink of CMOS transistors [16].

Fig. 6 (a) shows the power distribution network (PDN) consisting of all the power/ground planes from the voltage source to operating chips. Power is generated in VRM and delivered through electrical interconnections of PCB, package and chip. Finding ways to meet the desired voltage and current from the power source to destinations along the PDN is detrimental because failure in achieving

power integrity (PI) leads to various reliability problems such as incorrect switching of transistors, crosstalk from neighboring signals, and timing margin errors [17]. Decoupling capacitors (decaps) placed on the PDN allows the reliable power supply to the operating chips, thus improving the power integrity of hardware. As shown in Fig. 6 (b)-(c), the role of decap is analogous to that of water storage tanks, placed along the city, apartment, and household, that can provide water uninterruptedly and reliably. As if placing more water tanks can make the water supply more stable, placing more decaps can make power supply more reliable. However, because adding more decaps requires more space and is costly, optimally placement of decaps is important in terms of PI and cost/space-saving.

## A.2 PDN and Decap Models for Verification

**Unit-Cell Segmentation Method.** The segmentation method [18] is a simple and fast way to generate approximated electrical models. Because the analysis of the full electrical model using EM simulation is very time-consuming, we divided the full PDN model into smaller unit-cells and constructed the full PDN model using the unit-cell segmentation method. For fast simulation, we used equation-based python implemented segmentation method, illustrated in Fig. 7.

Segmentation method was used for generation of PDN model consisting of a chip layer and a package layer for verification as illustrated in Fig. 7 (a). The segmentation method was also used for objective evaluation of DPP. When a solution for DPP is made, decaps are placed on the corresponding ports on PDN using the segmentation method as illustrated in Fig. 7 (b).

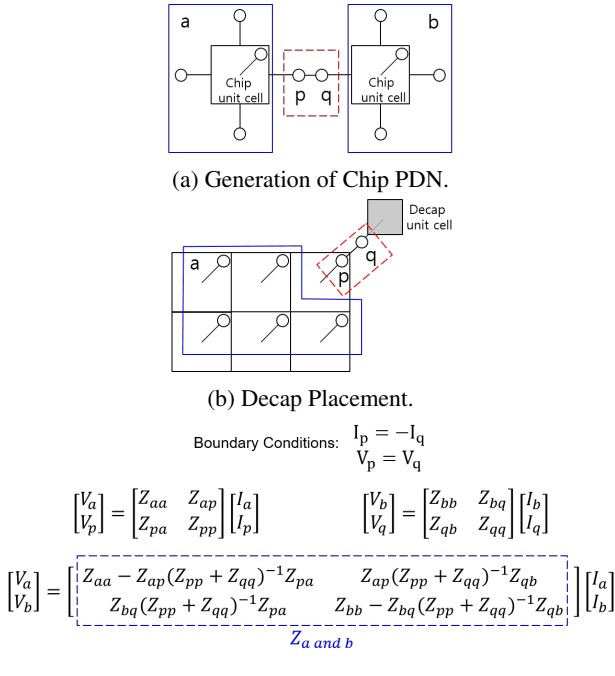

(a) Generation of Chip PDN.

(b) Decap Placement.

Boundary Conditions: $I_p = -I_q$
$V_p = V_q$

$$\begin{bmatrix} V_a \\ V_p \end{bmatrix} = \begin{bmatrix} Z_{aa} & Z_{ap} \\ Z_{pa} & Z_{pp} \end{bmatrix} \begin{bmatrix} I_a \\ I_p \end{bmatrix} \qquad \begin{bmatrix} V_b \\ V_q \end{bmatrix} = \begin{bmatrix} Z_{bb} & Z_{bq} \\ Z_{qb} & Z_{qq} \end{bmatrix} \begin{bmatrix} I_b \\ I_q \end{bmatrix}$$

$$\begin{bmatrix} V_a \\ V_b \end{bmatrix} = \underbrace{\begin{bmatrix} Z_{aa} - Z_{ap}(Z_{pp} + Z_{qq})^{-1}Z_{pa} & Z_{ap}(Z_{pp} + Z_{qq})^{-1}Z_{qb} \\ Z_{bq}(Z_{pp} + Z_{qq})^{-1}Z_{pa} & Z_{bb} - Z_{bq}(Z_{pp} + Z_{qq})^{-1}Z_{qb} \end{bmatrix}}_{Z_{a \, and \, b}} \begin{bmatrix} I_a \\ I_b \end{bmatrix}$$

(c) Segmentation Method].

Figure 7: Segmentation Method Implemented for PDN Generation and Decap Placement on PDN.

The PDN model we used for verification has a two-layer structure; a package layer at the bottom and a chip layer on top of it as illustrated in Fig. 8. The PDN was modeled through the unit-cell segmentation method. Package layer was composed of $40 \times 40$ package unit-cells and chip layer was composed of $10 \times 10$ (i.e, $N_{row} \times N_{col}$) chip unit-cells. Because the DPP benchmark places MOS type decaps, which are placed on chip, ports are only available on chip. Thus, we extracted $10 \times 10$ ports information from the chip layer. See Fig. 11 (a), illustrating the chip PDN divided into $10 \times 10$ units and each unit-cell numbered.

The electrical models of package and chip unit-cells that are used to build the PDN model for verification are described in Fig. 9. The chip layer is composed of $10 \times 10$ unit-cells, and the package layer is composed of $40 \times 40$ unit-cells using the segmentation method. The corresponding values of electrical parameters are listed in Table 3.

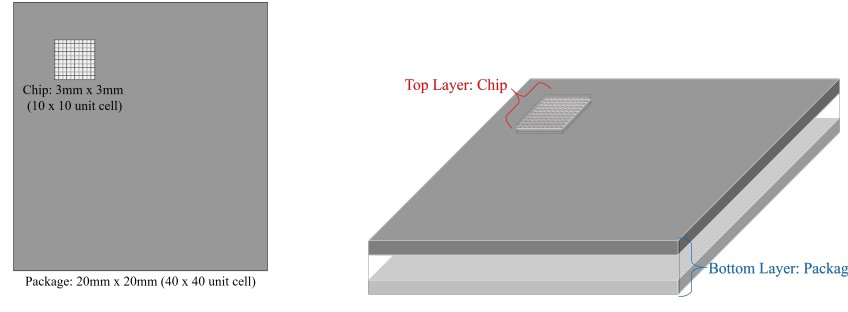

(a) Top-View of PDN model.    (b) Side-View of PDN model.

Figure 8: Top-view and Side-view of PDN Model used for Verification

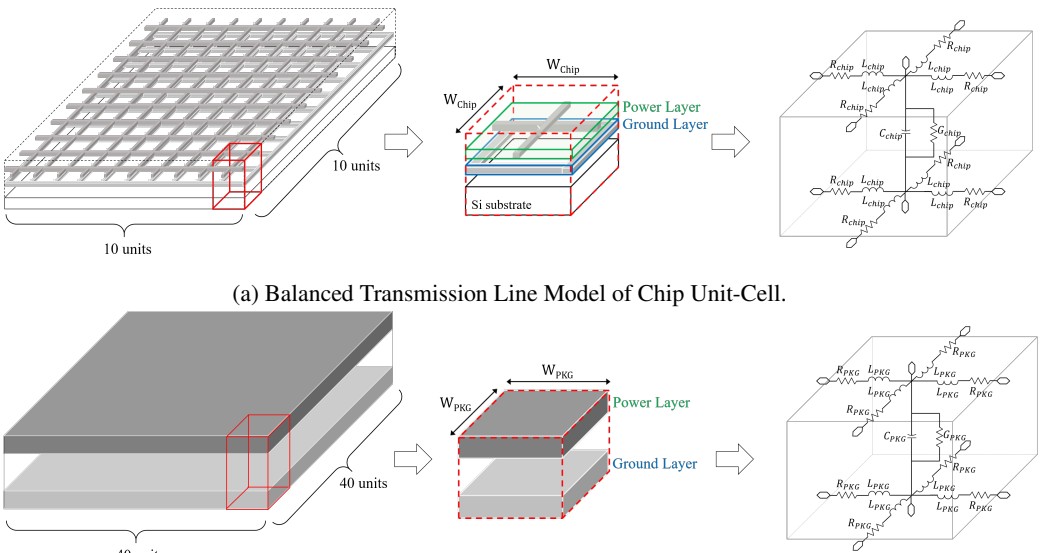

(a) Balanced Transmission Line Model of Chip Unit-Cell.

(b) Balanced Transmission Line Model of Package Unit-Cell.

Figure 9: Electrical Modeling of Chip and Package Unit-Cells for PDN Model generation.

Table 3: Width and Electrical Parameters for Chip and Package Unit-Cells used for Verification

| Unit-Cell Model | W | R | L | G | C |
|---|---|---|---|---|---|
| Chip | $300\mu$m | $0.26\,\Omega$ | 22pH | 1.2mS | 0.77pF |
| Package | 0.5mm | $0.093\,\Omega$ | 0.25nH | $5.4\mu$S | 0.045pF |

We implemented MOS type decap for verification. Decap model and its electrical parameters are shown in Fig. 10. As mentioned in Appendix A.1 Fig. 7 (b), the solution to DPP is evaluated using the segmentation method.

Note that these electrical parameters and PDN structures were used as a benchmark. For practical use of CDML, these PDN and decap models can be substituted by those of your interests.

### A.3    Input Problem PDN and Output Decap Placement Data Structure

Each unit-cell (i.e, port) of the PDN model described in Appendix A.2 is represented as a 3D vector composed of x-coordinate, y-coordinate and a number representing port state; 1 representing keep-out region, 2 representing a probing port and 0 for the rest. Total $10 \times 10$ (i.e, $N_{row} \times N_{col}$) 3D vectors

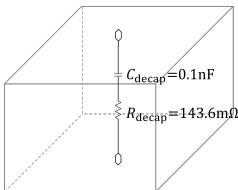

Figure 10: Decap Unit-Cell with the Electrical Parameters used for Verification.

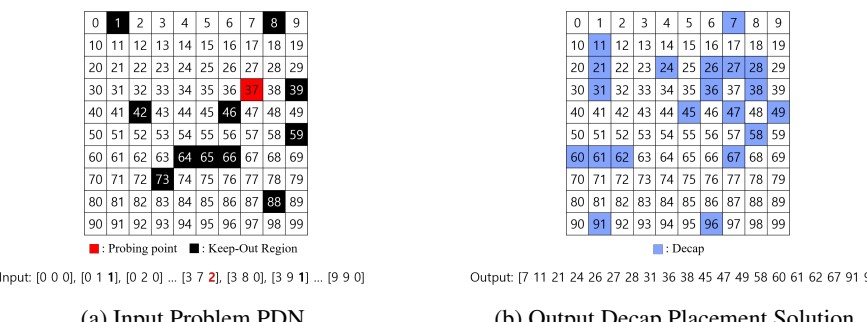

(a) Input Problem PDN.

(b) Output Decap Placement Solution.

Figure 11: Illustration of How the DPP problem is given as an Input and Decap Placement Solution is given as an Output.

represent the problem PDN. The solution to DPP is the placement of decaps. As illustrated in Fig. 11 (b), the solution is given as port numbers corresponding to each decap location.

## A.4 Random Problem Generation of DPP

To randomly generate decap placement problems (DPPs), $s_0 = \{p, \boldsymbol{ko}\}$, for training, test and validation, a probing index $p$ is selected randomly from a uniform distribution of $\{1, ..., N_{row} \times N_{col}\}$. Then keep-out region indices $ko$ are randomly selected through the following two stages: the number of keep-out regions $|ko|$ is randomly selected from a uniform distribution of $0 \sim 15$. Then, a vector containing indices of keep-out ports $\boldsymbol{ko}$ is generated by random selection from the uniform distribution of $\{1, ..., N_{row} \times N_{col}\}$. We generated 100 test problems and 100 validation problems for $10 \times 10$ PDN and 50 test problems and 50 validation problems for $15 \times 15$ PDN. We made sure the training, test, and validation problems do not overlap.

# B   Expert Label Collection

We used a genetic algorithm (GA) as the expert policy to collect expert guiding labels for imitation learning. GA is the most widely used search heuristic method for DPP [19, 20, 21, 22]. We devised our own GA for DPP, the objective of which is to find the placement of given number ($K$) of decaps on PDN with a probing port and 0-15 keep-out regions that best suppresses the impedance of the probing port.

**Notations.** $M$ is the number of samples to undergo an objective evaluation to give the best solution. The value of $M$ is defined by the size of population $P_0$ times the number of generation $G$. $K$ refers to the number of decaps to be placed. $P_{elite}$ is the number of elite population.

**Guiding Dataset.** To generate expert labels, guiding problems were generated in the same way test dataset was generated. We made sure the guiding data problems do not overlap with the test dataset problems. Also, we made sure each guiding problem does not overlap with each other. Each guiding data problem goes through the following process described in Fig. 12 to collect the corresponding expert label.

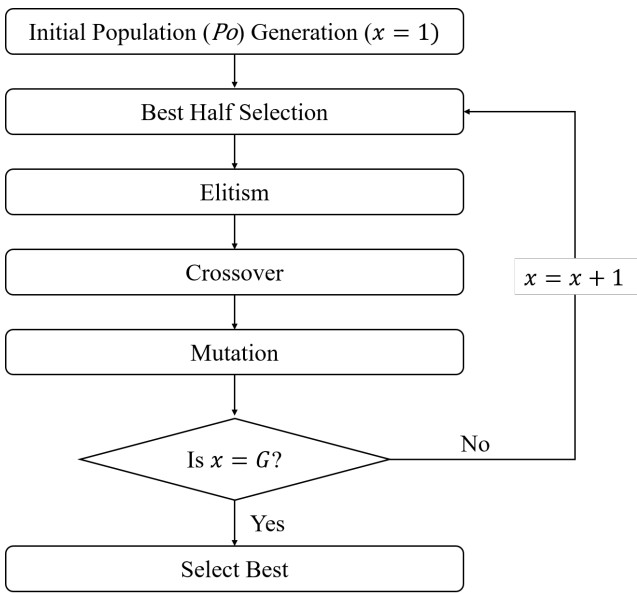

Figure 12: Process Flow of Genetic Algorithm for DPP.

**Population and Generation.** For GA $\{M = 100\}$ (*expert policy*), we fixed the size of population as $P_0 = 20$ and the number of generation as $G = 5$, which makes up total number of samples to be $M = P_0 \times G = 100$. Each solution in the initial population is generated randomly. As described in Fig. 11 (b), each solution consists of $K$ numbers, each representing a decap location on PDN. Note that each solution consists of random numbers from 0 to 99 except numbers corresponding to probing port and keep-out region locations.

Once the initial population is generated randomly, a new population is generated through elitism, crossover, and mutation. This whole process of generating a new population makes one generation; the Generation process is iterated for $G - 1$ times.

**Elitism.** Once initial population is formulated, the entire population undergoes objective evaluation and gets sorted in order of objective value. The size of elite population is pre-defined as $P_{elite} = 4$ for GA $\{M = 100\}$ (*expert policy*). That means the top 4 solutions in the population become the elite population and are kept for the next generation.

**Crossover.** Crossover is a process by which new population candidates are generated. Each solution of the current population including the elites is divided in half. Then, as described in Fig. 13 (c), half the solutions on the left and the other half on the right go through random crossover for $P_0$ times to generate a new population. If the elite population is available, $P_0 - P_{elite}$ random crossover takes place so that the total population size becomes $P_0$, including the elite population.

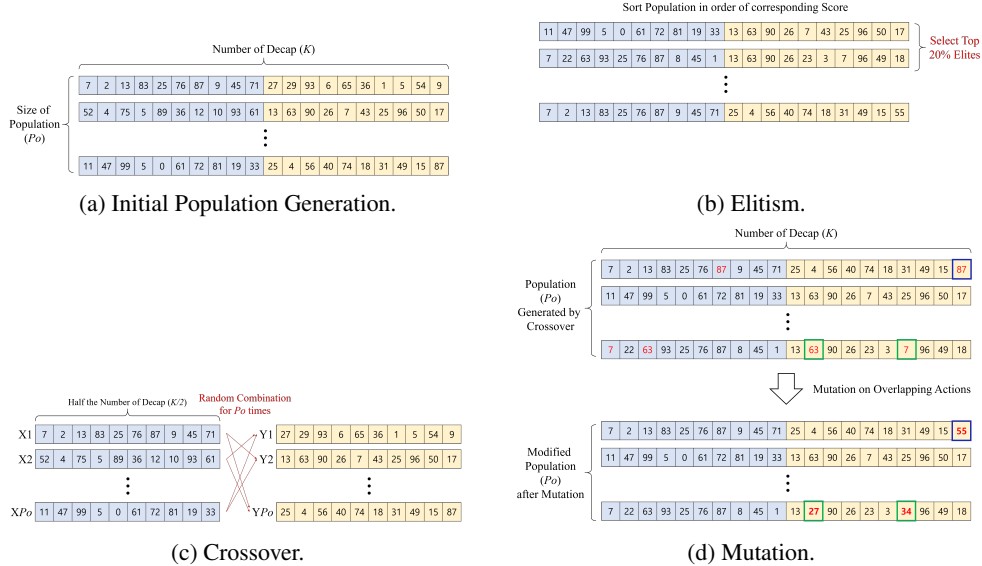

(a) Initial Population Generation.

(b) Elitism.

(c) Crossover.

(d) Mutation.

Figure 13: Illustration of each GA Operators used for DPP Guiding Data Generation.

**Mutation.** According to Fig. 13 (d), there may exist solutions with overlapping numbers after the random crossover. We replace the overlapping number with a randomly generated number, and we call this mutation.

**Select Best.** When $G$ is reached, the final population is evaluated by the performance metric. Then, a solution with the highest objective value becomes the final guiding solution for the given DPP.

The guiding problems and corresponding solutions generated as a result of GA are saved and used as guiding expert labels for imitation learning.

# C Details of Neural Architecture Design

Our neural architecture has the AM [10] with context modification. The AM is a transformer[23]-based encoder-decoder model designed to solve combinatorial optimization problems. We used conventional notations from transformer [23] and AM [10], including multi-head attention (MHA), feed forward (FF), query, key and value ($Q, K, V$). Because their terminologies are well organized, we tried to keep every notation as possible. In this paper, we focused on presenting the main differences between AM and our architecture. See [10] for detailed mechanism of AM.

## C.1 Change of Notations.

There are small revisions we made from [10]. In AM, TSP nodes are presented as $\boldsymbol{x}_i$, $i \in \{1, ..., N\}$, where $N$ refers to the number of TSP nodes. This paper uses $\boldsymbol{x}_p$ for node of the probing port, $\boldsymbol{x}_{1:|\boldsymbol{ko}|}$ for nodes of the keep-out regions and $\boldsymbol{x}_{1:\boldsymbol{d}}$ for nodes of the decap available place.

[10] denotes action as $\boldsymbol{\pi}$ (for representing permutation action), but we denoted action as $\boldsymbol{a}$.

In, [10], the notation, $\boldsymbol{h}^{(N)}$, refers to $N$ times MHA in encoder; we denoted this notation as $\boldsymbol{h}$ just for readability.

There are two additional notations: $\boldsymbol{c}_{(p)}$ is the probing context embedding from the probing port context network (PCN in section 3.2) and $\boldsymbol{c}_{a_{t-1}}$ is the recurrent context embedding from the recurrent context network (RCN in section 3.2) for $step = t$.

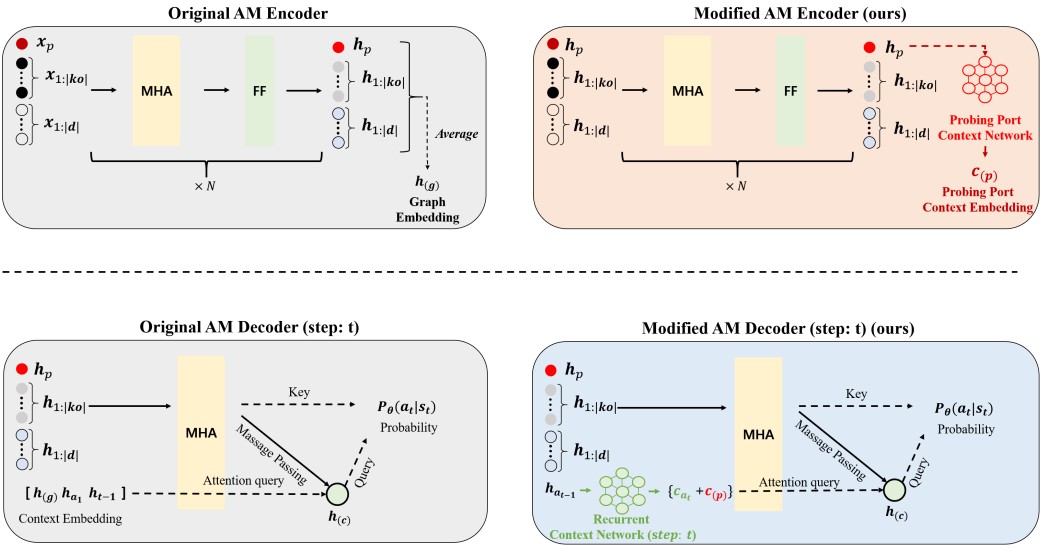

Figure 14: Overview of main difference between AM and modified version of AM.

## C.2 Highlight of modifications: Context Embedding.

The main difference between the AM and ours is the context embedding and is illustrated in Fig. 14.

AM's [10] context embedding is presented as follows:

$$\mathbf{h}_{(c)} = MHA([\mathbf{h}_{(g)}, \mathbf{h}_{a_{\tau-1}}, \mathbf{h}_{a_1}]) \tag{7}$$

**Context embedding of AM.** Since the AM was originally designed for TSP and its invariant problems, AM's context embedding is implemented for capturing the entire graph by taking the average of all node embedding, $\mathbf{h}_{(g)}$, state-transition with $\mathbf{h}_{a_{t-1}}$ and final destination with $\mathbf{h}_{a_1}$. Note that TSP is a routing problem, where it must return to the first node (i.e, destination node is first visited node).

**Context embedding of AM for DPP (AM-RL [15]).** [15] also used the AM for decap placement with modification of context embedding. [15] tried to add $\mathbf{h}^p$ to capture the location of probing port as follows:

$$\mathbf{h}_{(c)} = MHA([\mathbf{h}_{(g)}, \mathbf{h}_{a_{t-1}}, \mathbf{h}_p]) \tag{8}$$

**Context embedding of Ours.** We observed that $\mathbf{h}_{(g)}$ degrades the performance of the model for DPP. DPP is different from TSP; we need a new DPP-specific context embedding strategy. Therefore, we tried to focus on the probing port more than others by proposing the PCN. We excluded $\mathbf{h}_{(g)}$ and $\mathbf{h}_{a_1}$ from the context embedding and replaced them with our newly designed context embedding. Our context embedding is described as follows:

$$\mathbf{h}_{(c)} = MHA(\boldsymbol{c}_{(p)} + \mathbf{c}_{a_t}) \tag{9}$$

$$\mathbf{c}_{(p)} = \mathbf{MLP}_{PCN}(\mathbf{h}_p) \tag{10}$$

$$\mathbf{c}_{a_t} = \mathbf{MLP}_{RCN}(\mathbf{h}_{a_t}) \tag{11}$$

Note that both $\mathbf{MLP}_{PCN}$ and $\mathbf{MLP}_{RCN}$ are two-layer perceptron models with ReLU activation, where input and output dimensions are identical ($d = 128$ in all experiments).

## C.3  Calculation of Probability.

Probability calculations using $\boldsymbol{h}_{(c)}$, and $\boldsymbol{h}_i$, $i \in \{1, ..., N \times M\}$ in (11-14) are exactly identical to (5-8) in [10] except the masking mechanism in equation 13 and equation 14. Because [10] solves TSP, so they mask the previously selected actions by forcing $-\infty$ as compatibility $u_{(c)j}$. For DPP, we mask not only the previously selected actions $\boldsymbol{a}_{1:t-1}$ but also the probing port index $p$ and the keep-out region indices $\boldsymbol{ko}$; it is forbidden to choose the indices in current state $\boldsymbol{s_{t-1}} = \{p, \boldsymbol{ko}, \boldsymbol{a}_{1:t-1}\}$.

Query, key and value are computed by:

$$\boldsymbol{q}_c = W^Q \boldsymbol{h}_{(c)}, \boldsymbol{k}_i = W^K \boldsymbol{h}_i, \boldsymbol{v}_i = W^V \boldsymbol{h}_i \tag{12}$$

Note that $W^Q$, $W^K$ and $W^V$ are 128-to-128 linear projections.

After that, compatibility $u_{(c)j}$ is computed by the dot product of query and key, with masking mechanism (setting $-\infty$ not to select actions in $\boldsymbol{s}_{t-1}$).

$$u_{(c)j} = \begin{cases} \frac{\mathbf{q}_{(c)}^T \mathbf{k}_j}{\sqrt{128}} & \text{if } j \notin \boldsymbol{s_{t-1}} \\ -\infty & \text{otherwise} \end{cases} \tag{13}$$

The $tanh$ clipping is done following [24] and [10].

$$u_{(c)j} = \begin{cases} 10 \cdot \tanh\left(\frac{\mathbf{q}_{(c)}^T \mathbf{k}_j}{\sqrt{128}}\right) & \text{if } j \notin \boldsymbol{s_{t-1}} \\ -\infty & \text{otherwise.} \end{cases} \tag{14}$$

Finally, probability can be computed using softmax function as follows:

$$p_{\boldsymbol{\theta}}(a_t = i \mid \boldsymbol{s_{t-1}}) = \frac{e^{u_{(c)i}}}{\sum_j e^{u_{(c)j}}} \tag{15}$$

# D    Detailed Experimental Settings

This section provides detailed experimental settings for main experiments and ablation studies.

## D.1    Training Hyperparameters.

There are several hyperparameters for training; we tried to fix the hyperparameters as [10] did for showing their frameworks' practicality. We then provided several ablation studies on each hyperparameter to analyze how each component contributes to performance improvement.

Training hyperparameters are set to be identical to those presented in AM for TSP [10] except learning rate, unsupervised regularization rate $\lambda$, the number of expert data $N$, number of action permutation transformed data per expert data $P$ and batch size $B$.

Table 4: Hyperparameter setting for training model.

| Hyperparameter | Value |
|---|---|
| learning rate | 0.00001 |
| $\lambda$ | $8 \times 10^{32}$ |
| $N$ | 1000 |
| $P$ | 3 |
| $B$ | 1000 |

## D.2    Implementation of ML Baselines.

There are two main ML baselines, Arb-RL [14] and AM-RL [15].

**Arb-RL.** Arb-RL is a PointerNet-based DPP solver proposed by [14]. However, reproducible source code was not available. Therefore, we implemented the Arb-RL following the implementation of [24] [1] and paper of [14]. We set the training step $1,600$ with batchsize $B = 100$ that makes total $160,000$ PI simulation.

**Arb-IL.** Arb-IL is an imitation learning version of Arb-RL trained by our training data. We set $N = 2000$, $B = 1000$ for training Arb-IL.

**AM-RL.** AM-RL is a AM-based DPP solver proposed by [15]. We reproduced AM-RL by following implementation of [10] [2] and paper of [15]. We set the training step $1,600$ with batchsize $B = 100$ that makes total $160,000$ PI simulation.

**AM-IL.** AM-IL is an imitation learning version of AM-RL trained by our training data. For experiments in Table 1, we set $N = 2000$ and $B = 1000$ for training. For ablation study, we mainly ablate $N$, when $N = 100$ we set $B = 100$. Here is the training sample complexity (the number of PI simulations during training) of each ML baselines and CDML:

Table 5: Training sample complexity of ML baselines and CDML.

| Methods | The Number of PI simulations for Training |
|---|---|
| Arb-RL | 160,000 |
| AM-RL | 160,000 |
| Arb-IL $\{N = 2000\}$ | 200,000 ($N = 2000$, $M = 100$ from GA expert) |
| AM-IL $\{N = 2000\}$ | 200,000 ($N = 2000$, $M = 100$ from GA expert) |
| **CDML** $\{N = 100\}$ (ours) | 10,000 ($N = 100$, $M = 100$ from GA expert) |
| **CDML** $\{N = 1000\}$ (ours) | 100,000 ($N = 1000$, $M = 100$ from GA expert) |

During the inference phase, each learned model produces a greedy solution from their policy (i.e., $M = 1$) following [10].

---

[1]https://github.com/pemami4911/neural-combinatorial-rl-pytorch
[2]https://github.com/wouterkool/attention-learn-to-route

### D.3 Implementation of Meta-Heuristic Baselines.

**Genetic Algorithm (GA).** GA $\{M = 100\}$ and GA $\{M = 500\}$ are implemented as baselines. For detailed procedures and operators used for GA, see Appendix.B. GA $\{M = 100\}$ is the expert policy used to generate expert data for imitation learning in CDML. For GA $\{M = 100\}$, the size of population, $P_0$, is 20, number of generation, $G$, is 5 and elite population, $P_{elite}$, is 4. For GA $\{M = 500\}$, $P_0$ is 50, $G$ is 10 and $P_{elite}$ is 10.

**Random Search (RS).** The random search method generates $M$ random samples for a given problem and selects the best sample with the highest objective value.

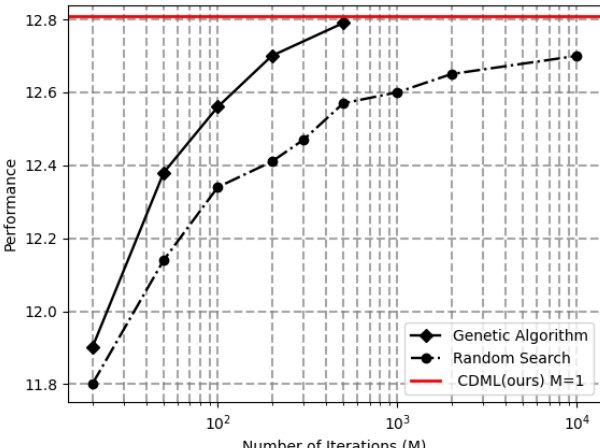

Figure 15: Performance of GA and RS with varying number of iterations ($M$) in comparison to CDML at $M = 1$.

Fig. 15 shows the performance of GA and RS depending on the number of iterations ($M$). The performance was measured by taking the average of 100 test data solved by each method at each $M$. GA outperformed RS at every $M$, and the performance increased with increasing $M$ for both methods. However, the gradient of performance increment decreased with increasing $M$. On the other hand, our CDML showed higher performance than GA$\{M = 100\}$ and RS $\{M = 10,000\}$ with a single inference $M = 1$.

# E    Experimental Results in terms of Power Integrity

The objective of DPP is to suppress impedance of the probing port as much as possible over a specified frequency range and is measured by the objective metric, $Obj := \sum_{f \in F} (Z_{initial}(f) - Z_{final}(f)) \cdot \frac{1\text{GHz}}{f}$. Performance of CDML was evaluated in comparison to GA $\{M = 100\}$ (*expert policy*), GA $\{M = 500\}$, RS $\{M = 10,000\}$, AM-RL and AM-IL on unseen 100 PDN cases. Each method was asked to place 20 decaps $(K = 20)$ on each test.

## E.1    Impedance Suppression Plots

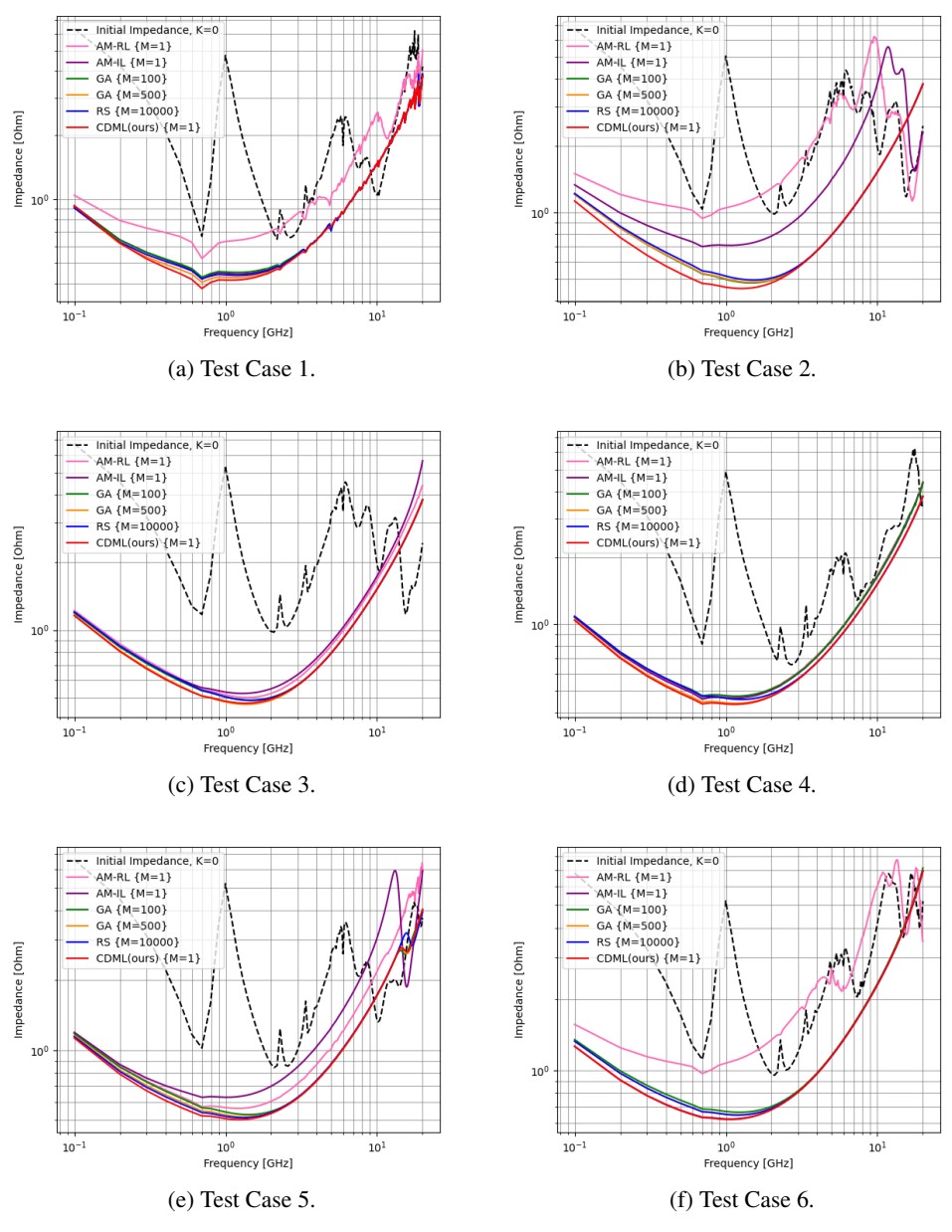

Figure 16: Impedance suppressed by each method, GA $\{M = 100\}$ (*expert policy*), GA $\{M = 500\}$, RS $\{M = 10,000\}$, AM-RL , AM-IL and CDML (Ours) for 6 example PDN cases out of 100 test dataset. (The lower the better.)

## E.2 Decap Placement Tendency Analysis

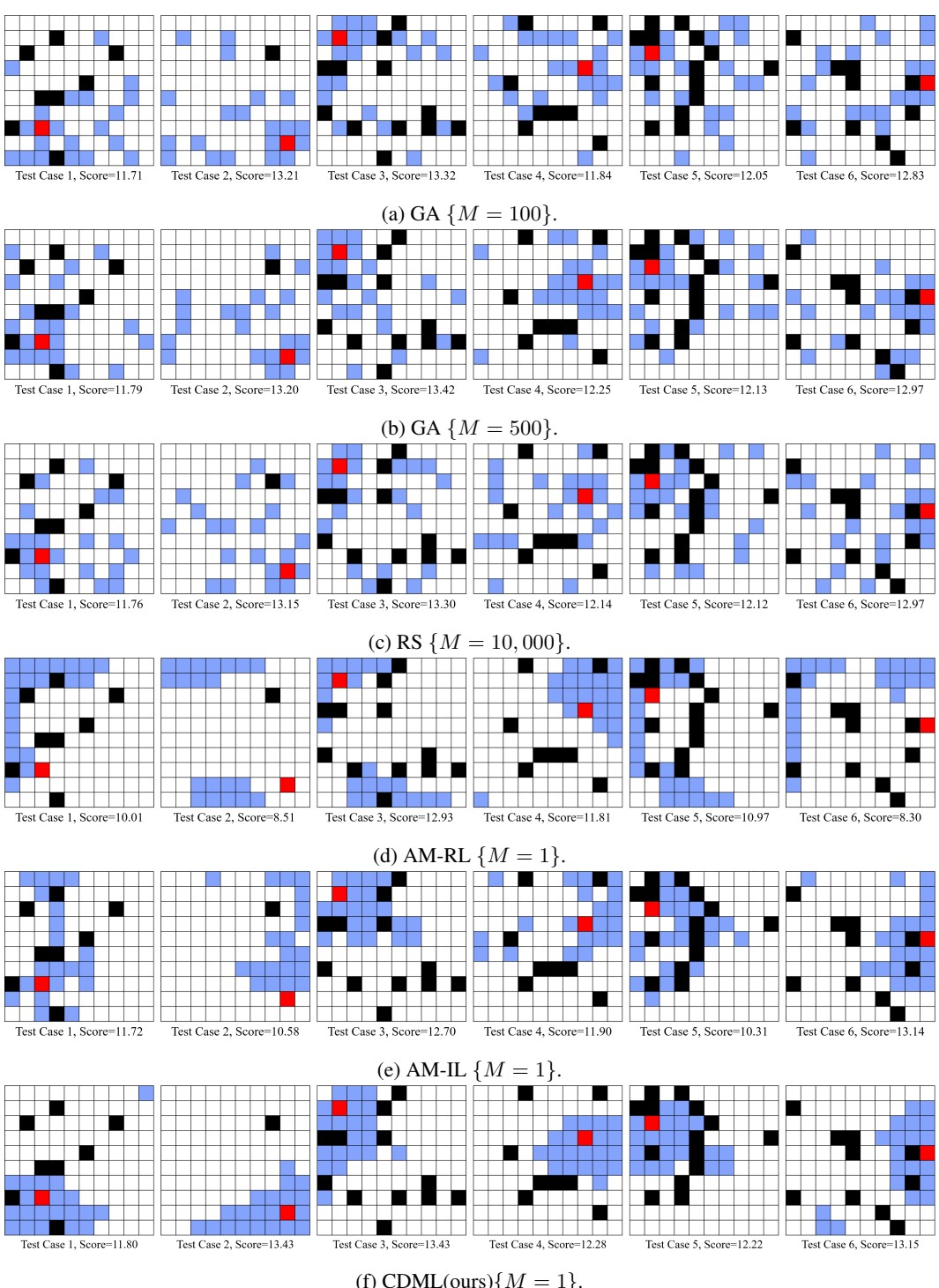

Figure 17: Corresponding decap placement solutions to Fig. 16 by each method. Red represents probing port, black represents keep-out ports and blue represents decap locations.

Fig. 17 shows the decap placement solutions of 6 PDN cases plotted in Fig. 16. The solutions by the search-heuristic methods, GA and RS, tend to be scattered while the solutions by learning-based methods, AM-RL, AM-IL and CDML, are clustered. Since search-heuristic methods are based on

random generations, they do not show clear tendency. On the other hand, learning based methods are based on a policy so that they have distinct tendency in placing decaps.

The role of placing decaps in hardware design is to decouple loop inductance of PDN. In terms of PI, analysis of loop inductance is critical, but at the same time, is complex [25]. The loop inductance distribution of PDN highly depends on various design parameters such as the location of probing port, spacing between power/ground, size of PDN, and hierarchical layout of PDN [26]. When human experts place decaps on PDN, there are too many domain rules to consider. On the other hand, CDML understands the PDN structure and its electrical properties by data-driven learning. According to Fig. 17, CDML tends to place decaps near the probing port, which is a well-known expert rule in the PI domain.

# F Further Ablation Study

This section reports ablation studies on action permutation invariance and hyperparameters $N$ (number of guiding samples), $\lambda$ (weight of self-distillation loss term), and $P$ (number of permutation transformed labels).

## F.1 Ablation Study on $N$

$N$ is the number of expert labels generated by the expert policy, GA $\{M = 100\}$. We ablate $N \in \{100, 500, 1000, 2000\}$ with fixed $P = 3$ and $\lambda = 8$ and compare to AM-IL baseline for all $N$. As shown in Table 6, CDML with $N = 2000$ gives the best performance and CDML outperforms AM-IL for all $N$ variations. Performance of AM-IL is saturated at $N > 500$ while the performance of CDML continuously increases with the increase of $N$.

Table 6: Ablation study on $N$ for CDML ($P = 3, \lambda = 8$) and AM-IL.

|  | Validation Score |
| --- | --- |
| AM-IL $\{N = 100\}$ | 11.60 |
| CDML (ours) $\{N = 100\}$ | **12.98** |
| AM-IL $\{N = 500\}$ | 12.37 |
| CDML (ours) $\{N = 500\}$ | **12.99** |
| AM-IL $\{N = 1000\}$ | 12.23 |
| CDML (ours) $\{N = 1000\}$ | **13.09** |
| AM-IL $\{N = 2000\}$ | 12.32 |
| CDML (ours) $\{N = 2000\}$ | **13.13** |

## F.2 Ablation Study on $\lambda$

$\lambda$ refers to the weight of self-distillation loss term $L_{Self}$, in the collaborative learning loss $\mathcal{L} := \mathcal{L}_{Expert} + \lambda \mathcal{L}_{Self}$. To set $\lambda \times L_U$ be $0.1 \sim 1$, we first multiplied $10^{32}$ to $\lambda$ because the probability of a specific solution is extremely small. Then, we ablated for $\lambda \in \{1, 2, 4, 6, 7, 8, 9, 10\}$ ($10^{32}$ is omitted) with fixed $N = 100$ and $P = 3$. For every $\lambda$, it prevents overfitting of the model in comparison to the baselines trained only with $L_{Expert}$ (see Fig. 18). According to the Table 8, $\lambda = 8$ gives the best validation scores.

Table 7: Ablation study of $\lambda$ on fixed $P = 3$ and $N = 100$.

| $\lambda\,(\times 10^{32})$ | Validation Score |
|---|---|
| 1 | 12.96 |
| 2 | 12.96 |
| 4 | 12.94 |
| 6 | 12.96 |
| 7 | 12.98 |
| 8 | **12.98** |
| 9 | 12.97 |
| 10 | 12.96 |
| Only IL, $\lambda = 0$ | 12.97 |

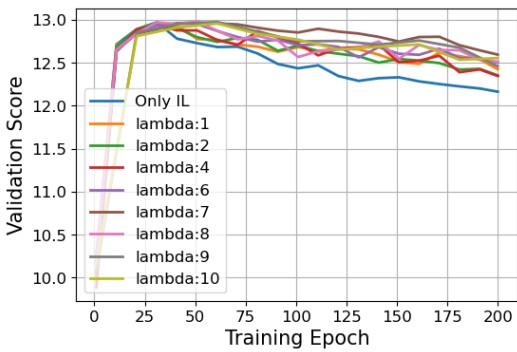

Figure 18: Validation graph of $\lambda \in \{1, 2, 4, 6, 7, 8, 9, 10\}$ on fixed $P = 3$ and $N = 100$.

### F.3 Ablation Study on $P$

$P$ is the number of permutation transformed labels per each expert label used for expert distillation imitation learning. We ablate $P \in \{3, 5, 7\}$ with fixed $N = 100$ and $\lambda = 8$ and compared collaborative distillation (i.e., both expert and self-distillation) to only expert distillation training case. As shown in Table 8, $P = 3$ with {Expert distillation + Self-distillation} give best performances. For every $P$, {Expert distillation + Self-distillation } gives the better performances, indicating self-distillation scheme well prevents overfitting of training process for sparse dataset.

Table 8: Ablation study on $P$ with and without unsupervised loss term.

|  | Validation Score |
| --- | --- |
| Expert distillation $\{P = 3\}$ | 12.97 |
| + Self- distillation $\{\lambda = 8\}$ | **12.98** |
| Expert distillation $\{P = 5\}$ | 12.95 |
| + Self- distillation $\{\lambda = 8\}$ | **12.95** |
| Expert distillation $\{P = 7\}$ | 12.93 |
| + Self- distillation $\{\lambda = 8\}$ | **12.95** |

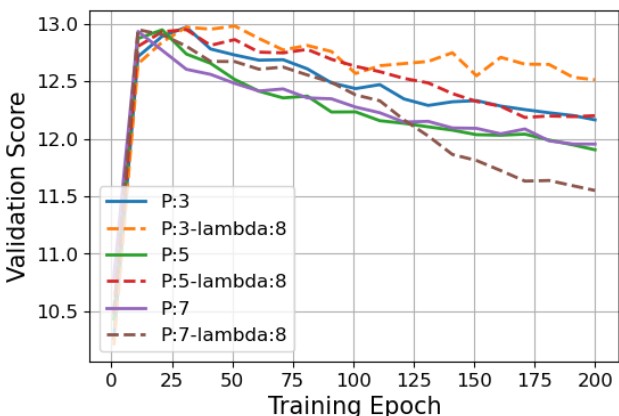

Figure 19: Validation score of $P$ ablation with and without Unsupervised Loss term.

