# OpenReview forum: "Collaborative symmetricity exploitation for offline learning of hardware design solver"
_NeurIPS.cc/2022/Workshop/Offline_RL — Offline RL Workshop NeurIPS 2022_

### Official Review · Reviewer_2VZt · 2022-10-08
**A BC method for hardware design with action trajectory permutation invariance.**

**Rating:** 7
**Confidence:** 4

**Review:**

The paper proposes a BC-based system for decoupling capacitor placement. In this problem for any action sequence, the result is independent of the order in which these actions are taken. The proposed method is to train the BC policy on data augmented via random permutation of actions in the trajectory. Additionally, a self-consistency loss enforces that the current policy outputs the same distribution for different permutations of the state sequence. The method is shown to improve over vanilla BC training as well as search-based methods. Overall, the paper presents a potentially valuable contribution to hardware design.

---

### Official Review · Reviewer_BaXk · 2022-10-19
**Application poses an interesting problem for offline RL**

**Rating:** 7
**Confidence:** 2

**Review:**

This paper introduces a framework for offline RL which is applied to the decoupling capacitor placement problem. The framework exploits symmetries to efficiently learn a policy. The policy is parameterized by a task condition, thus requires some generalization ability. I found the paper to be quite clear despite my lack of expertise in the application area, and the problem well motivated. It appears to be a nice practical use of RL that utlizes offline expert data. The method itself is also interesting and fairly well-explained. One thing that is not clear to me is how the self-generated data is produced. Is this part of the method conducted online? If so, it would be good to highlight that this is not a purely offline method. However, its benefits appear quite clear from the experiments and this is not meant to be a critique. I was particularly glad to see the ablation study and experimental analysis of offline dataset size.

Overall, I could see this paper inviting an interesting discussion in the workshop.

Some points
- Perhaps this work is relevant: Elise van der Pol, Daniel Worrall, Herke van Hoof, Frans Oliehoek, Max Welling. MDP Homomorphic Networks: Group Symmetries in Reinforcement Learning, NeurIPS 2020.
- I would be interested in seeing how this approach could be generalized to a wider family of problems, and how that family might be defined